

# Development and validation of a 36-gene sequencing assay for hereditary cancer risk assessment

Valentina S. Vysotskaia[1,*], Gregory J. Hogan[1,*], Genevieve M. Gould[1,*], Xin Wang[1], Alex D. Robertson[1,5], Kevin R. Haas[1], Mark R. Theilmann[1], Lindsay Spurka[1], Peter V. Grauman[1], Henry H. Lai[1], Diana Jeon[1], Genevieve Haliburton[1], Matt Leggett[2], Clement S. Chu[1], Kevin Iori[1], Jared R. Maguire[1], Kaylene Ready[3], Eric A. Evans[1], Hyunseok P. Kang[4] and Imran S. Haque[1,6]

[1] Research and Development Department, Counsyl, Inc, South San Francisco, CA, United States
[2] Project Management Department, Counsyl, Inc, South San Francisco, CA, United States
[3] Medical Affairs Department, Counsyl, Inc, South San Francisco, CA, United States
[4] Clinical Laboratory, Counsyl, Inc, South San Francisco, California, United States
[5] Current affiliation: Color Genomics, Inc., Burlingame, CA, United States
[6] Current affiliation: Freenome, Inc., South San Francisco, CA, United States
[*] These authors contributed equally to this work.

Corresponding author
Hyunseok P. Kang,
peter@counsyl.com

## ABSTRACT

The past two decades have brought many important advances in our understanding of the hereditary susceptibility to cancer. Numerous studies have provided convincing evidence that identification of germline mutations associated with hereditary cancer syndromes can lead to reductions in morbidity and mortality through targeted risk management options. Additionally, advances in gene sequencing technology now permit the development of multigene hereditary cancer testing panels. Here, we describe the 2016 revision of the Counsyl Inherited Cancer Screen for detecting single-nucleotide variants (SNVs), short insertions and deletions (indels), and copy number variants (CNVs) in 36 genes associated with an elevated risk for breast, ovarian, colorectal, gastric, endometrial, pancreatic, thyroid, prostate, melanoma, and neuroendocrine cancers. To determine test accuracy and reproducibility, we performed a rigorous analytical validation across 341 samples, including 118 cell lines and 223 patient samples. The screen achieved 100% test sensitivity across different mutation types, with high specificity and 100% concordance with conventional Sanger sequencing and multiplex ligation-dependent probe amplification (MLPA). We also demonstrated the screen's high intra-run and inter-run reproducibility and robust performance on blood and saliva specimens. Furthermore, we showed that pathogenic Alu element insertions can be accurately detected by our test. Overall, the validation in our clinical laboratory demonstrated the analytical performance required for collecting and reporting genetic information related to risk of developing hereditary cancers.

# INTRODUCTION

Tremendous advances in our knowledge of evaluating and treating patients with germline mutations associated with hereditary cancer syndromes have been realized in the past two decades. Multiple studies demonstrate the feasibility and clinical utility of genetic testing (*Norton et al., 2007*; *Domchek et al., 2010*; *Kurian et al., 2014*; *Lynce & Isaacs, 2016*). Most importantly, studies have provided convincing evidence that identification of hereditary cancer syndromes can lead to reductions in morbidity and mortality through targeted risk management options. For example, for unaffected women who carry a *BRCA1* or *BRCA2* mutation, risk-reducing salpingo-oophorectomy results in a significant reduction in all-cause mortality (3% vs. 10%; hazard ratio [HR] 0.40; 95% CI [0.26–0.6]), breast cancer-specific mortality (2% vs. 6%; HR 0.44; 95% CI [0.26–0.76]), and ovarian cancer–specific mortality (0.4 vs. 3%; HR 0.21; 95% CI [0.06–0.8]) when compared with carriers who chose not to undergo this procedure (*Domchek et al., 2010*).

Until recently, the traditional approach for germline testing was to test for a mutation in a single gene or a limited panel of genes (syndrome-based testing) using Sanger sequencing (*Sanger, Nicklen & Coulson, 1977*), quantitative PCR (*Barrois et al., 2004*), and MLPA (*Hogervorst et al., 2003*). With advances in next-generation DNA sequencing (NGS) technology and bioinformatics analysis, testing of multiple genes simultaneously (panel-based testing) at a cost comparable to traditional testing is possible. NGS-based, multigene panels of 25–79 genes have been developed and are offered by several clinical diagnostic laboratories (*Easton et al., 2015*; *Kurian & Ford, 2015*; *Lynce & Isaacs, 2016*). Panel-based testing has proven to provide improved diagnostic yield (*Rehm, 2013*; *Castéra et al., 2014*; *Cragun et al., 2014*; *Kurian et al., 2014*; *LaDuca et al., 2014*; *Lincoln et al., 2015*; *Minion et al., 2015*). Among clinic-based studies that collectively assessed more than 10,000 patients who tested negative for *BRCA1/2* mutations, mutation prevalence in non-*BRCA* genes ranged from 4% to 16% (*Castéra et al., 2014*; *LaDuca et al., 2014*; *Kurian et al., 2014*; *Maxwell et al., 2015*; *Tung et al., 2015*). Some mutations were clinically unexpected (e.g., a *MSH6* mutation, consistent with Lynch syndrome, was found in a patient with triple-negative breast cancer) (*Kurian et al., 2014*), prompting calls for a change in screening and prevention recommendations.

Published validation studies demonstrate high analytical concordance between results from NGS and the traditional Sanger method for detection of sequence level variations (single-nucleotide variants, small deletions and insertions) (*Bosdet et al., 2013*; *Chong et al., 2014*; *Judkins et al., 2015*; *Lincoln et al., 2015*; *Strom et al., 2015*). However, detection of exon-level copy number variations and larger indels might be relatively challenging for NGS (*Lincoln et al., 2015*). To address this concern, some laboratories complement NGS with microarrays (*Chong et al., 2014*). Other laboratories achieve high accuracy of NGS-based copy number variation and indel detection using sophisticated bioinformatics pipelines (*Lincoln et al., 2015*; *Kang et al., 2016*; *Schenkel et al., 2016*). Although this is encouraging, it is important to consider the potential limitations of NGS for detection of larger insertions/deletions (indels) and copy number variants (CNVs, also known as deletions and duplications or large rearrangements). Samples with technically challenging classes of mutations should be included in analytical validation.

Here, we describe the development and validation of the 2016 revision of the Counsyl Inherited Cancer Screen, an NGS-based test to identify single nucleotide variants (SNVs), indels, and copy number variants in 36 genes associated with an elevated risk for breast, ovarian, colorectal, gastric, endometrial, pancreatic, thyroid, prostate, melanoma, and neuroendocrine cancers. To evaluate analytical performance of the test and ensure quality of results, we followed the American College of Medical Genetics and Genomics (ACMG) guidelines for analytical validation of NGS methods (*Rehm et al., 2013*). The validation study included both well-characterized cell lines ($N = 118$) and de-identified patient samples ($N = 223$) with clinically relevant variants.

## MATERIALS AND METHODS

### Institutional review board approval

The protocol for this study was approved by Western Institutional Review Board (IRB number 1145639) and complied with the Health Insurance Portability and Accountability Act (HIPAA). The information associated with patient samples was de-identified in accordance with the HIPAA Privacy Rule. A waiver of informed consent was requested and approved by the IRB.

### Multigene panel design

Thirty-six genes associated with hereditary forms of cancer, including breast, ovarian, colorectal, gastric, endometrial, pancreatic, thyroid, prostate, melanoma, and neuroendocrine, were selected for development of the Counsyl Inherited Cancer Screen panel. The genes are: *APC, ATM, BARD1, BMPR1A, BRCA1, BRCA2, BRIP1, CDH1, CDK4, CDKN2A, CHEK2, EPCAM, GREM1, MEN1, MLH1, MRE11A, MSH2, MSH6, MUTYH, NBN, PALB2, PMS2, POLD1, POLE, PTEN, RAD50, RAD51C, RAD51D, RET, SDHA, SDHB, SDHC, SMAD4, STK11, TP53,* and *VHL* (Table 1). Twenty nine of the 36 genes were specifically included due to the availability of patient management guidelines by NCCN or other professional societies. Further details regarding the panel are available in Table S1.

The selected genes are tested for SNVs, indels, and CNVs throughout coding exons and 20 bp of flanking intronic sequences. Additionally, known deleterious variants outside the coding regions are sequenced. In *EPCAM*, only large deletions that include exon 9 are reported as these mutations are known to silence the *MSH2* gene (*Tutlewska, Lubinski & Kurzawski, 2013*). In *GREM1*, specific pathogenic duplications in the promoter, which are commonly associated with individuals of Ashkenazi Jewish descent, are covered. Specifically, the screen targets the three most common promoter duplications in *GREM1* (coordinates with respect to GRCh37/hg19 reference assembly):

- chr15:32,964,939-33,004,759 (40kb)
- chr15:32,986,220-33,002,449 (16kb)
- chr15:32,975,886-33,033,276 (57kb).

For *PMS2*, exons 11–15 are excluded from the reportable region of interest (ROI) because of high similarity between this portion of *PMS2* and its highly homologous pseudogene *PMS2CL*. In *RET*, exon 1 is not sequenced due to high guanine-cytosine (GC) content.

**Table 1** List of 36 genes included in the inherited cancer screen panel.

| Gene | Transcript:exon sequenced | SNV/indel reportable ROI, bp | Variants reported |
| --- | --- | --- | --- |
| APC | NM_000038: 2–16 | 9,433 | SNVs, indels, CNVs |
| ATM | NM_000051: 2–63 | 11,853 | SNVs, indels, CNVs |
| BARD1 | NM_000465: 1–11 | 2,776 | SNVs, indels, CNVs |
| BMPR1A | NM_004329: 3–13 | 2,046 | SNVs, indels, CNVs |
| BRCA1 | NM_007294: 2–23 | 7,351 | SNVs, indels, CNVs |
| BRCA2 | NM_000059: 2–27 | 11,652 | SNVs, indels, CNVs |
| BRIP1 | NM_032043: 2–20 | 4,556 | SNVs, indels, CNVs |
| CDH1 | NM_004360: 1–16 | 3,350 | SNVs, indels, CNVs |
| CDK4 | NM_000075: 2–8 | 1,229 | SNVs, indels, CNVs |
| CDKN2A | NM_000077: 1–3 | 1,343 | SNVs, indels, CNVs |
| CHEK2 | NM_007194: 2–15 | 2,199 | SNVs, indels, CNVs |
| EPCAM | NM_002354: 9 | | CNVs |
| GREM1 | NM_013372: upstream duplications | | CNVs |
| MEN1 | NM_000244: 2–10 | 2,306 | SNVs, indels, CNVs |
| MLH1 | NM_000249: 1–19 | 3,295 | SNVs, indels, CNVs |
| MRE11A | NM_005591: 2–20 | 2,897 | SNVs, indels, CNVs |
| MSH2 | NM_00025: 1–16 | 3,692 | SNVs, indels, CNVs |
| MSH6 | NM_000179: 1–10 | 4,566 | SNVs, indels, CNVs |
| MUTYH | NM_001048171: 1–16 | 2,321 | SNVs, indels, CNVs |
| NBN | NM_002485: 1–16 | 2,905 | SNVs, indels, CNVs |
| PALB2 | NM_024675: 1–13 | 4,090 | SNVs, indels, CNVs |
| PMS2 | NM_000535: 1–10 | 1,649 | SNVs, indels, CNVs |
| POLD1 | NM_001256849: 2–27 | 4,435 | SNVs, indels, CNVs |
| POLE | NM_006231: 1–49 | 8,823 | SNVs, indels, CNVs |
| PTEN | NM_000314: 1–9 | 1,866 | SNVs, indels, CNVs |
| RAD50 | NM_005732: 1–25 | 4,944 | SNVs, indels, CNVs |
| RAD51C | NM_058216: 1–9 | 1,509 | SNVs, indels, CNVs |
| RAD51D | NM_002878: 1–10 | 1,862 | SNVs, indels, CNVs |
| RET | NM_020975: 2–20 | 4,167 | SNVs, indels, CNVs |
| SDHA | NM_004168: 1–15 | 2,606 | SNVs, indels, CNVs |
| SDHB | NM_003000: 1–8 | 1,188 | SNVs, indels, CNVs |
| SDHC | NM_003001: 1–6 | 864 | SNVs, indels, CNVs |
| SMAD4 | NM_005359: 2–12 | 2,148 | SNVs, indels, CNVs |
| STK11 | NM_000455: 1–9 | 1,717 | SNVs, indels, CNVs |
| TP53 | NM_000546: 2–11 | 1,818 | SNVs, indels, CNVs |
| VHL | NM_000551: 1–3 | 789 | SNVs, indels, CNVs |

## Next Generation DNA Sequencing

Our application of next-generation DNA sequencing is performed as described previously (*Kang et al., 2016*). Briefly, DNA from a patient's blood or saliva sample is isolated, quantified by a dye-based fluorescence assay, and then fragmented to 200–1,000 bp by sonication. The fragmented DNA is converted to a sequencing library by end repair, A-tailing, and adapter ligation. Samples are then amplified by PCR with barcoded primers,

multiplexed, and subjected to hybrid capture-based enrichment with 40-mer oligonucleotides (Integrated DNA Technologies, Coral, IL, USA) complementary to targeted regions. Next generation sequencing of the selected targets is performed with sequencing-by-synthesis on the Illumina HiSeq 2500 instrument to a mean sequencing depth of $\sim$650$\times$. All target nucleotides are required to be covered with a minimum depth of 20 reads.

## Bioinformatics processing

Sequencing reads are aligned to the hg19 human reference genome using the BWA-MEM algorithm (*Li, 2013*). Single-nucleotide variants and short indels are identified and genotyped using GATK 1.6 and FreeBayes (*McKenna et al., 2010*; *Garrison & Marth, 2012*). The calling algorithm for copy number variants is described below. All SNVs, indels, and large deletions/duplications within the reportable range are analyzed and classified by the method described in the section "Variant Classification". All reportable calls are reviewed by licensed clinical laboratory personnel.

## CNV calling algorithm

Copy number variants for samples are determined by inspecting the number of mapped reads observed at targeted positions in the genome across samples in a flowcell lane. Our method is based upon previous successful approaches applying hidden Markov models (HMMs) to exome sequencing data (*Plagnol et al., 2012*) with modifications presented below that have been optimized for accurate resolution of CNVs based on the particulars of the sequencing technology. As sequencing depth is linearly proportional to the number of copies of the genome at that position, we construct a statistical model for the likelihood of observing a given number of mapped reads $d_{i,j}$ at a given genomic position $i$ for sample $j$ with copy number $c_{i,j}$.

The expected number of reads is dependent upon 3 factors: the average depth for that targeted location across samples $\mu_i$, the average depth for that particular sample across targeted positions $\mu_j$, and the local copy number of the sample's genome at that targeted position. These are first determined by finding the median depth at targeted region across all $N_s$ samples in an analyzed flowcell lane

$$\mu_i = \frac{\sum_j d_{i,j}}{N_s}$$

then the sample dependent factor $\mu_j$ is found by taking the median across all $N_p$ positions in genome after normalizing for the expected number of reads at each position

$$\mu_j = \frac{\sum_i d_{i,j}/\mu_i}{N_p}.$$

Combining these factors the observed data are modeled by the negative binomial distribution

$$p(d_{i,j}|c_{i,j}) = NegBinom(d_{i,j}|\mu = c_{i,j}\mu_i\mu_j, r = r_i).$$

This characterization has been found to accurately model the observed number of reads from previous targeted sequencing experiments (*Anders & Huber, 2010*).

In the negative binomial model, the variance parameter $r_i$ accounts for regions of the genome where sequencing depth is observed to follow idealized Poisson statistics in the limit that $r \to \infty$ and regions that are excessively noisy with respect to observed number of reads when $r \to 0$.

$r_i$ may be estimated as

$$r_i = \frac{\mu_i^2}{Var_j[d_{i,j}] - \mu_i}$$

which is found to closely model the empirical distribution over several orders of magnitude in read depth.

Because duplications and deletions will simultaneously impact the expected depth of all genomic positions encompassing the variant, depth data from spatially adjacent positions are correlated. We leverage the HMM to account for this correlation. The HMM's state transition probabilities between wild-type and copy-number-variant are parameterized by matching the average length of such variations observed in human population (*Sudmant et al., 2015*) through setting $p_{CNV \to WT} = 1/6,200$ between each subsequent base-pair and a prior on the frequency of such variations

$$\frac{p_{WT \to CNV}}{p_{CNV \to WT}} = p_{CNV}.$$

The prior $p_{CNV} = 0.001$ was determined by balancing the thresholds for confident calling and retesting of calls to achieve the desired sensitivity and specificity, and the prior was set independently of this validation.

Detecting CNVs using this probabilistic framework invokes the Viterbi algorithm (*Korn et al., 2008*) to determine the most likely number of copies at every targeted region within a sample. Any contiguous regions of duplication or deletion produce a reported variant, and the confidence of that call is determined by aggregating the posterior probability of the call $\sum_{i \in CNV} p(c_{i,j} \neq 2)$ not being wildtype over the called region.

All called copy number variants are inspected for quality by human review. To avoid false positives, all patient samples with a called CNV are tested a second time, starting with a new DNA extraction and including library preparation, sequencing and bioinformatic analysis. Samples that emit low confidence called variants are additionally rerun to resolve a confident genotype.

### Detection of Alu insertions

Alu positives were detected by looking for Alu sequences in reads overlapping with Alu insertion positions. All insertions were only tested at positions where the sequence had been previously confirmed by Sanger sequencing. At the site of an Alu insertion, the Alu sequence is soft-clipped by BWA alignment. These soft-clipped reads were compiled; duplicate reads were discarded; and the remaining reads with sequences matching the known Alu sequence at this site were tallied. Sites with at least three unique reads matching the Alu sequence were called as Alu positive.

## Pre- and post-sequencing quality metrics

To ensure the quality of the results obtained from the assay, 27 different review checkpoints (Table S2) were developed. Ancillary quality-control metrics are computed on the sequencing output and used to exclude and re-run failed samples, and include the fraction of sample contamination (<5%), extent of GC bias, read quality (percent Q30 bases per Illumina specifications), depth of coverage (per base minimum coverage $\geq 20\times$ and mean coverage of >250×), and region of interest (ROI) coverage (100%). Calls that do not meet criteria listed in Table S2 are set to "no-call." To ensure clinical calling accuracy, all calls and no-calls for potentially deleterious variants, and all calls for variants of unknown significance are manually reviewed by laboratory personnel and are subject to override if warranted, based on a pre-established protocol.

## Variant classification

Variants are classified using multiple lines of evidence according to the ACMG Standards and Guidelines for the Interpretation of Sequence Variants (*American College of Medical Genetics and Genomics, 2015*; *Richards et al., 2015*). Variants that are known or likely to be pathogenic are reported; patients and providers have an option to have variants of uncertain significance reported as well. Final variant classifications are regularly uploaded to ClinVar (*Landrum et al., 2014*), a peer-reviewed database created with a goal of improving variant interpretation consistency between laboratories.

## Statistical analysis

Variant calls were defined as true positive for variants identified by the Counsyl Inherited Cancer Screen and by independent testing (the 1000 Genomes Project or MLPA/Sanger data), false positive for variants identified by the Counsyl test but not by the independent data, and false negative for variants identified by the independent data but not by the Counsyl test. To estimate true negatives, we counted polymorphic sites (positions at which we observed non-reference bases in any sample) with concordant negative results across all considered samples. No-calls were censored from the analysis. As no-calls have the potential to introduce clinically relevant false negatives, we separately examined the no-calls containing potentially deleterious alleles by treating no-calls as homozygous reference and comparing to the 1000 Genomes calls. We found all no-calls when treated as homozygous reference were concordant with the exception that one comparison was inconclusive due to low allele balance in both our data and the exome data from the 1000 Genomes Project (Table S8).

Validation metrics were defined as: Accuracy = (TP +TN) / (TP +FP +TN +FN); Sensitivity = TP / (TP +FN); Specificity = TN / (TN +FP); FDR = FP / (TP +FP), where TP = true positives, TN = true negatives, FP = false positives, FN =false negatives, and FDR = false discovery rate. The confidence intervals (CIs) were calculated by the method of Clopper and Pearson (*Clopper & Pearson, 1934*). To estimate reproducibility within and between runs, the ratio of concordant calls to total calls was calculated.

**Table 2  Source of samples and reference data used in validation.**

| Measures | Variant type | Test samples | Reference data |
|---|---|---|---|
| Accuracy, sensitivity, specificity | SNV, indel | 101 Coriell cell line samples | 1000 Genomes project exomes |
| | | 2 Coriell cell lines with specific mutations | Coriell data |
| | | 2 NIBSC samples | NIBSC reference data |
| | | 82 mutation-positive patient samples | Orthogonal confirmation by Sanger |
| Accuracy, sensitivity, specificity | CNV | 5 NIBSC samples | NIBSC reference data |
| | | 44 CNV-positive patient samples | Orthogonal confirmation by MLPA |
| Intra-run reproducibility | SNV, indel, CNV | 8 Genome-in-a-Bottle (GiaB) cell line samples | |
| | | 13 patient samples | |
| Inter-run reproducibility | SNV, indel, CNV | 8 GiaB cell line samples | |
| | | 84 patient samples | |

## Study samples

The validation sample set comprised (a) 111 genomic DNA reference materials purchased from the Coriell Cell Repositories (Camden, NJ) (Table S3), (b) *MLH1/MSH2* exon copy number reference panel from the National Institute for Biological Standards and Control ($N = 7$) (Table S4), and (c) 223 deidentified patient samples used for MLPA- and Sanger-based confirmation (Tables 2 and S4).

The validation set included samples with reference data for SNVs and indels (the 1000 Genomes Project), a broad range of indels (both short ≤10 bp and long >10 bp) characterized by Sanger sequencing, homopolymer-associated variants, Alu element insertions, and both single- and multi-exon copy-number variants characterized by MLPA (Table 3). Validation material was derived from cell lines, blood, and saliva samples. Collectively, the validation set provides broad coverage of known relevant types of genomic variation across the reportable region of the test (Tables 3 and S5). A list of the validation samples from Sanger and MLPA confirmation is provided in Table S4.

## RESULTS

### Test description

We developed an NGS-based test that interrogates 36 genes associated with hereditary cancer risk (Table 1). The majority of the 36 genes were selected based on the availability of patient management guidelines developed by NCCN or other professional societies. The reportable region of interest (ROI) of the test is 124,245 bp representing coding exons, intron boundaries and non-exonic mutation-containing regions (Table 1). The wet lab protocols and reagents are carefully optimized to ensure 100% coverage of targeted base pairs at an average depth of 650 reads and a minimal depth of 20 reads sufficient for robust detection of multiple classes of genomic alterations: single-nucleotide variations, indels, and copy number variations.

**Table 3** Variants in validation study.

| Variant type | Deletion/insertion size | Total (unique) number of variants | |
|---|---|---|---|
| | | Reference data | Orthogonal confirmation |
| SNV | | 5,182 (425) | |
| Indel | Indels ≤10 bp | | 57 (29) |
| | Indels >10 bp | | 19 (15) |
| Alu insertion | | | 7 (4) |
| CNV | Single-exon deletions or duplications | 3 (3) | 10 (9) |
| | Multiple exon deletions or duplications | 2 (2) | 35 (27) |

## Validation approach

Several regulations, including the Clinical Laboratory Improvement Act of 1988 (CLIA), the ACMG guidelines for analytical validation of NGS methods (*Rehm et al., 2013*), as well as various quality standards for diagnostic laboratories require rigorous analytical validation of panel tests for clinical use. In contrast to diagnostic assays for a single gene or a limited panel of genes (syndrome-based testing), analytical validation of a NGS-based test assaying 36 genes for multiple types of genomic alterations is a complex task. To address this challenge, we developed a representative validation approach with reference samples selected to cover variant and specimen variability that may affect test accuracy and reproducibility for clinical use.

To measure the accuracy of SNV and indel detection, we tested samples from the 1000 Genomes Projects with reference data for SNVs and indels in all 36 genes. Testing on the 1000 Genomes Project samples allows us to assess the ability to call commonly observed variant types and the ability to test calling in regions that may be difficult for NGS due to considerable sequence homology (e.g., *CHEK2*, *SDHA*, and *PMS2*) or low complexity (homopolymer runs). However, the 1000 Genomes reference samples provide limited validation for technically challenging variants like CNVs, larger indels, and Alu insertions. To build a collection of reference material to test such challenging variants, we identified relevant patient samples tested with a previous version of the Counsyl test (a 24-gene panel) and orthogonally confirmed each of the positive samples by either Sanger or MLPA. Using these cohorts of reference samples (e.g., samples with CNVs), we could then assess call accuracy for each type of technically challenging variant on this newly designed 36-gene panel. Finally, to validate test reproducibility, we examined SNV, indel, and CNV calls in cell line and patient (blood and saliva) samples processed independently in several batches (inter-run reproducibility) or tested repeatedly in the same batch (intra-run reproducibility).

## Analytical validation for SNVs and indels

The analytical validation of the Inherited Cancer Screen was performed according to ACMG guidelines (*Rehm et al., 2013*) and in accordance with the requirements of CLIA for medical laboratories. SNV and indel detection was examined on a 101-sample validation set consisting of reference samples from the 1000 Genomes Project with known SNV and indel sites across the targeted regions (Tables 2 and S5). Counsyl sequence data for 36 genes were

**Table 4  Performance of Counsyl Inherited Cancer Screen for SNVs and indels.**

|  | Counsyl test | 1000 Genomes Project data | | Results (95% confidence interval) |
|---|---|---|---|---|
|  |  | Variant present | Variant not present |  |
| SNV & Indel | Variant detected | 5,182 true positives | 0 false positives | 100% accuracy (99.991–100%) |
|  |  |  |  | 100% sensitivity (99.93–100%) |
|  | Variant not detected | 0 false negatives | 37,743 true negatives | 100% specificity (99.990–100%) |
|  |  |  |  | 0% FDR (0-0.0007%) |

**Notes.**
Validation metrics were defined as: Accuracy = (TP +TN)/(TP +FP +TN +FN); Sensitivity = TP/(TP +FN); Specificity = TN/(TN +FP); FDR = FP/(TP +FP). For true negative calculations, all polymorphic positions (positions at which we observed non-reference bases in any sample) across all samples were considered.

compared to reference data obtained from the 1000 Genomes Projects. Out of 42,925 total calls validated, 18 calls were discordant between Counsyl and the 1000 Genomes Project (Table S6). One of the 18 discordances was a potential false positive variant call, identified as a variant by the Counsyl test, but identified as reference by the 1000 Genomes Project. The remaining 17 calls were potential false negative variants identified by the 1000 Genomes Project, but not by the Counsyl test. Manual review of the 1000 Genomes reference data for each of the discordant sites using the Integrated Genomics Viewer (IGV) (*Robinson et al., 2011*; *Thorvaldsdóttir, Robinson & Mesirov, 2013*) found that a large portion of the discordant calls came from hard-to-sequence (e.g., highly homologous *SDHA* gene) or low-coverage regions, which is a reported limitation in the 1000 Genomes Project (*1000 Genomes Project Consortium et al., 2012*). With that in mind, each of the discordant sites was subjected to Sanger sequencing as an independent testing method and the data from Sanger sequencing supported all 18 of Counsyl's calls as true positives or true negatives (Table S6).

Analytical validation results of Counsyl's test for SNV and indel detection is presented in Table 4. Counsyl's test identified 5,182 true positive calls, 37,743 true negative calls, and no false positive nor false negative calls, resulting in 100% sensitivity (95% CI [100%–99.93%]), 100% specificity (95% CI [100%–99.99%]), and 0% FDR (95% CI [0–0.0007%]) of the test for detecting SNVs and indels.

## Validation of challenging variants
### *CNVs*
To assess the accuracy of CNV detection, we measured the concordance between Counsyl's test results on 44 blood and saliva samples with CNV positives confirmed by MLPA (N = 43) or Sanger (N = 1) (Tables 2 and S4b). For one CNV positive sample (Counsyl_147), Sanger sequencing was used for orthogonal confirmation. MLPA analysis of this sample failed to identify the partial deletion of coding sequence in the terminal exon of *APC* because the deletion was relatively small and fell between the MLPA probes (Table S4b). For the patient sample Counsyl_128, two duplications affecting exons 8–9 of *EPCAM* and exons 1–16 of *MSH2* were detected and confirmed by MLPA. Additionally, 5 NIBSC reference samples with known CNVs in the *MLH1* and *MSH2* genes were included in the validation. Among the 49 tested samples (a total of 50 CNVs), 13 had a single-exon

**Table 5  Performance of Counsyl Inherited Cancer Screen for indels and CNVs.**

| | Counsyl test | Sanger or MLPA reference data | | Results (95% confidence interval) |
|---|---|---|---|---|
| | | **Variant present** | **Variant not present** | |
| Indel | Variant detected | 76 true positives | 0 false positives | 100% accuracy (99.88–100%) |
| | | | | 100% sensitivity (95–100%) |
| | Variant not detected | 0 false negatives | 3,040 true negatives | 100% specificity (99.88–100%) |
| | | | | 0% FDR (0–5%) |
| CNV | Variant detected | 50 true positives | 0 false positives | 100% accuracy (99.5–100%) |
| | | | | 100% sensitivity (93–100%) |
| | Variant not detected | 0 false negatives | 685 true negatives | 100% specificity (99.5–100%) |
| | | | | 0% FDR (0–7.1%) |

**Notes.**
Validation metrics were defined as: Accuracy = (TP +TN)/(TP +FP +TN +FN); Sensitivity = TP/(TP +FN); Specificity = TN/(TN +FP); FDR = FP/(TP +FP). For indels, true negatives defined as the number of homozygous reference calls made at sites for which an alternative variant was observed in at least one sample in the cohort. For CNVs, true negatives defined as the number of genes assigned the reference copy number in the CNV validation cohort, and the summation included only genes for which a known CNV positive was tested ($N = 15$ genes with a CNV positive).

deletion or duplication, which can be technically challenging for a NGS-based assay (Table 3).

As shown in Table 5, we detected all 50 CNVs, including 13 single-exon events, demonstrating the high sensitivity of the assay (100%; 95% CI [100%–93%]). Furthermore, no additional CNV calls were made in the 49-sample cohort, resulting in 100% specificity (Table 5).

## Challenging indels

To measure accuracy for detecting indels, we built a cohort ($N = 82$) of patient samples with variants of a range of sizes, including both short ($\leq 10$ bp) and the more technically challenging long ($>10$ bp) deletions or insertions (Tables 3 and S4a). These samples were identified using a previous version of the Counsyl test (a 24-gene panel) and orthogonally confirmed by Sanger. We then tested these samples with the newly developed 36-gene panel and confirmed all of the expected indel calls; no false-positives nor false-negatives were observed in the 36-gene panel results (Table 5).

## Alu insertions

Alu elements represent a special class of insertions and are known to be clinically important (*Belancio, Roy-Engel & Deininger, 2010*). Alu insertions have been reported in *ATM*, *BRCA1, BRCA2,* and *BRIP1* (*Belancio, Roy-Engel & Deininger, 2010*; *Kennemer et al., 2016*), including known examples of Alu insertion founder mutations (e.g., c.156_157insAlu in *BRCA2* exon 3 in Portuguese populations) (*Peixoto et al., 2014*). Accurate detection of Alu insertions is challenging, especially for traditional Sanger sequencing where longer Alu-containing alleles are usually out-competed during PCR (*De Brakeleer et al., 2013*). To test the sensitivity of our assay and bioinformatics pipeline for Alu insertion detection, we included 7 positive cases (Portuguese founder mutation in exon 3 of *BRCA2,* Alu insertion in *BRCA2* exon 25 and intronic Alu insertions in *ATM* and *MSH6*) in our validation study

**Table 6** List of Alu insertions confirmed in validation.

| Sample ID | Gene | Variant description |
|---|---|---|
| Counsyl 24 | *ATM* | Intron 54–55, NM_000051.3: c.8010+13_8010+14insAlu |
| Counsyl 25 | *ATM* | Intron 54–55, NM_000051.3: c.8010+13_8010+14insAlu |
| Counsyl 26 | *ATM* | Intron 54–55, NM_000051.3: c.8010+13_8010+14insAlu |
| Counsyl 27 | *BRCA2* | Exon 3, NM_000059.3: c.156_157insAlu |
| Counsyl 28 | *BRCA2* | Exon 3, NM_000059.3: c.156_157insAlu |
| Counsyl 85 | *BRCA2* | Exon 25, NM_000059.3:c.930_931insAlu |
| Counsyl 84 | *MSH6* | Intron 2–3, NM_000179: c.458-19_458-18insAlu |

(Table 6). We confirmed that the Alu insertions identified by the Counsyl Inherited Cancer Screen were also detected by Sanger sequencing.

**Reproducibility**

In addition to establishing the test's analytical sensitivity and specificity, Counsyl's Inherited Cancer Screen was validated for intra- and inter-run call reproducibility. Intra-run reproducibility of SNV and indel calls was established by testing eight cell lines and 13 blood or saliva samples in 2–3 replicates in the same batch, split across sequencer lanes. Inter-run reproducibility was validated by testing eight cell lines and 84 patient blood or saliva samples in 2–3 different batches run by two operators, on different instruments and on different dates (Table S7a). Concordance between replicates was >99.99%, with just one discordant call at a known benign homopolymer site in an intron of *ATM* (Table S7a).

For CNVs, intra-run and inter-run reproducibility was established using the Coriell sample NA14626 with a duplication of *BRCA1* exon 12 (Table S7b). Concordance between eight replicates was 100%, with no differences between inter- and intra-run replicates observed.

# DISCUSSION

The evidence base for genetic testing, counseling, risk assessment and management for hereditary cancer syndromes is rapidly evolving. The expansion of knowledge regarding cancer-risk associated genes and advances in gene sequencing technology now permit the development of multigene hereditary cancer testing panels. Recently, we have expanded the Counsyl Inherited Cancer Screen to 36 genes known to impact inherited risks for ten important cancers: breast, ovarian, colorectal, gastric, endometrial, pancreatic, thyroid, prostate, melanoma, and neuroendocrine. The 36-gene panel is fully customizable; however the genes on the panel can be divided into two groups: those genes with clinical utility particularly for the unaffected population and those genes with undefined clinical utility. Clinical utility is defined as genes with established cancer risks and risk management guidelines developed by professional societies such as the National Comprehensive Cancer Network. At the time of publication, 29 genes have clear clinical utility (Table S1). In a clinical setting and in a test report, cancer risks and risk management options for those genes with clinical utility are described in detail. Cancer risks are often provided as a range to reflect the fact that the exact risk for any one individual cannot be precisely known.

Including cancer risks and risk management options on the test report also provides the opportunity to inform patients and providers of the variation in risk and appropriate management for different genes. For example, while *BRCA1* mutations are associated with a 50–85% lifetime risk of breast cancer and consideration of prophylactic mastectomy may be appropriate, *ATM* mutations are associated with more moderate risks that may warrant screening with a breast magnetic resonance imaging (MRI) but not necessarily consideration of prophylactic mastectomy.

Accurate detection of clinically relevant genomic alterations in the targeted genes is critical and requires the interrogation of coding exons as well as selected non-coding regions with known pathogenic mutations. Furthermore, robust detection of a broad range of clinically relevant genomic alterations in routine clinical specimens, such as blood and saliva, is also required for a clinical-grade test. To address these challenges, we developed a clinical-grade, targeted NGS test for 36 genes. We carefully optimized and validated the probe design and NGS-based workflow using reference cell lines and clinical samples. We performed a comprehensive validation study and did not identify any false positives or false negatives. High sensitivity, specificity, accuracy and call reproducibility were observed across homozygous and heterozygous SNVs, indels, and CNVs, including technically challenging variants, such as single- and multi-exon deletions/duplications ($N = 50$), >10 bp indels ($N = 19$) and Alu insertions ($N = 7$). For patients with two heterozygous mutations, which could be clinically relevant for recessive diseases (e.g., MYH-associated polyposis), we are able to phase nearby mutations and demonstrate compound heterozygosity

Although some NGS validation studies report a higher false positive rate and require orthogonal confirmation of positive calls (*Chong et al., 2014*; *Mu et al., 2016*), high sensitivity and specificity consistent with this report have been achieved in similar studies, both in our laboratory (*Kang et al., 2016*) and in other laboratories (*Bosdet et al., 2013*; *Judkins et al., 2015*; *Lincoln et al., 2015*; *Strom et al., 2015*). No false negatives were observed in our study, corroborating previous reports of high analytic accuracy of NGS relative to Sanger sequencing (99.965%) (*Beck et al., 2016*). However, another recent publication uses data from 20,000 NGS panel tests performed in a clinical setting (Ambry Genetics, Aliso Viejo, CA) to claim the necessity of Sanger confirmation of variants detected by NGS (*Mu et al., 2016*). This study observed a 99/7845 (1.3%) false positive rate and concluded that Sanger confirmation is needed to maintain high accuracy, particularly in difficult-to-sequence regions. In contrast to other work in the field, Mu et al. state that it was impossible with their pipeline to reach a zero false negative rate when filtering NGS variant calls for a zero false positive rate. For example, the *MSH2*:c.942+3A>T variant, which falls at the end of a stretch of 27 adenines, was missed by Mu et al. in 5 of 6 patients when they tuned their false positive rate to zero.

The results presented here support the high accuracy for NGS calls, including challenging variants in hard-to-sequence regions, and demonstrate that the requirement for secondary confirmation is a property of each particular NGS pipeline, not a generic property of all NGS protocols. The *MSH2*:c.942+3A>T variant, highlighted as difficult in the Mu et al. publication, was included and correctly called in our validation data. Indeed, our cell line

and patient validation cohorts included 3,421 pathogenic and nonpathogenic variants (Table S5) in the gene set that exhibited false positives in Mu et al.'s study; for all 3,421 variants, we observed 100% analytical concordance with reference (1000 Genomes) and orthogonal confirmation (Sanger/MLPA) data.

The high accuracy reported here underlines the importance of using metrics beyond simple base and variant call quality to assess NGS variant calls. Table S2 shows the comprehensive set of metrics by which we assess each variant call. As one example, information on read directionality ("strand bias LOD") is incorporated into our pipeline, and would have eliminated many of the false positives encountered by Mu et al. (in particular, the *MSH2* homopolymer site) without sacrificing sensitivity. Finally, the call review process described here includes visual inspection of all potentially deleterious calls.

For copy number variants, the low throughput of non-NGS-based CNV analysis methods combined with the low prevalence of CNVs makes it difficult to assess CNV calling sensitivity with precision. While in principle orthogonal testing of all negative CNV calls using MLPA, qPCR, or microarrays may uncover additional samples with copy number variants, this would constitute a large discovery effort with low probability of discovering a false negative. The development of a set of reference samples with a diverse deeply-characterized collection of copy number variants (analogous to the efforts of the Genome in a Bottle project) would be a great benefit to laboratory validation procedures.

In conclusion, we developed a 36-gene sequencing test for hereditary cancer risk assessment. We assessed test performance across a broad range of genomic alteration types and clinical specimen properties to support clinical use. We confirmed high analytical sensitivity and specificity in this validation study consisting of 5,315 variants, including many technically challenging classes. The test is now offered by Counsyl's laboratory, which is CLIA certified (05D1102604), CAP accredited (7519776), and NYS permitted (8535).

### Funding
The authors received no funding for this work.

### Competing Interests
All authors, except Alex D. Robertson and Imran S. Haque, are current employees of Counsyl, Inc. Alex is a former employee of Counsyl and is currently employed by Color Genomics, Inc. Imran is also a former employee of Counsyl and is currently employed by Freenome, Inc.

### Author Contributions
- Valentina S. Vysotskaia conceived and designed the experiments, analyzed the data, wrote the paper, prepared figures and/or tables, reviewed drafts of the paper.
- Gregory J. Hogan and Genevieve M. Gould conceived and designed the experiments, analyzed the data, contributed reagents/materials/analysis tools, wrote the paper, prepared figures and/or tables, reviewed drafts of the paper.

- Xin Wang analyzed the data, contributed reagents/materials/analysis tools, reviewed drafts of the paper.
- Alex D. Robertson, Genevieve Haliburton, Matt Leggett, Clement S. Chu, Kevin Iori and Jared R. Maguire contributed reagents/materials/analysis tools, reviewed drafts of the paper.
- Kevin R. Haas contributed reagents/materials/analysis tools, wrote the paper, reviewed drafts of the paper.
- Mark R. Theilmann and Lindsay Spurka performed the experiments, analyzed the data, contributed reagents/materials/analysis tools, reviewed drafts of the paper.
- Peter V. Grauman analyzed the data, contributed reagents/materials/analysis tools, wrote the paper, reviewed drafts of the paper.
- Henry H. Lai and Diana Jeon performed the experiments, contributed reagents/materials/analysis tools, reviewed drafts of the paper.
- Kaylene Ready contributed reagents/materials/analysis tools, wrote the paper, prepared figures and/or tables, reviewed drafts of the paper.
- Eric A. Evans wrote the paper, reviewed drafts of the paper.
- Hyunseok P. Kang and Imran S. Haque wrote the paper, prepared figures and/or tables, reviewed drafts of the paper.

## Human Ethics

The following information was supplied relating to ethical approvals (i.e., approving body and any reference numbers):

Western Institutional Review Board (IRB number 1145639).

## Data Availability

ClinVar: https://www.ncbi.nlm.nih.gov/clinvar/submitters/320494.

## Supplemental Information

Supplemental information for this article can be found online at http://dx.doi.org/10.7717/peerj.3046#supplemental-information.

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

## FURTHER READING

**Antoniou AC, Casadei S, Heikkinen T, Barrowdale D, Pylkäs K, Roberts J, Lee A, Subramanian D, De Leeneer K, Fostira F, Tomiak E, Neuhausen SL, Teo ZL, Khan S, Aittomäki K, Moilanen JS, Turnbull C, Seal S, Mannermaa A, Kallioniemi A, Lindeman GJ, Buys SS, Andrulis IL, Radice P, Tondini C, Manoukian S, Toland AE, Miron P, Weitzel JN, Domchek SM, Poppe B, Claes KBM, Yannoukakos D, Concannon P, Bernstein JL, James PA, Easton DF, Goldgar DE, Hopper JL, Rahman N, Peterlongo P, Nevanlinna H, King M-C, Couch FJ, Southey MC, Winqvist R, Foulkes WD, Tischkowitz M. 2014.** Breast-cancer risk in families with mutations in PALB2. *New England Journal of Medicine* **371(6)**:497–506 DOI 10.1056/NEJMoa1400382.

**Apostolou P, Fostira F. 2013.** Hereditary breast cancer: the era of new susceptibility genes. *BioMed Research International* **2013**:747318 DOI 10.1155/2013/747318.

**Bartkova J, Tommiska J, Oplustilova L, Aaltonen K, Tamminen A, Heikkinen T, Mistrik M, Aittomäki K, Blomqvist C, Heikkilä P, Lukas J, Nevanlinna H, Bartek J. 2008.** Aberrations of the MRE11-RAD50-NBS1 DNA damage sensor complex in human breast cancer: MRE11 as a candidate familial cancer-predisposing gene. *Molecular Oncology* **2(4)**:296–316 DOI 10.1016/j.molonc.2008.09.007.

**Bellido F, Pineda M, Aiza G, Valdés-Mas R, Navarro M, Puente DA, Pons T, González S, Iglesias S, Darder E, Piñol V, Soto JL, Valencia A, Blanco I, Urioste M, Brunet J, Lázaro C, Capellá G, Puente XS, Valle L. 2016.** POLE and POLD1 mutations in 529 kindred with familial colorectal cancer and/or polyposis: review of reported cases and recommendations for genetic testing and surveillance. *Genetics in Medicine* **18(4)**:325–332 DOI 10.1038/gim.2015.75.

**Damiola F, Pertesi M, Oliver J, Le Calvez-Kelm F, Voegele C, Young EL, Robinot N, Forey N, Durand G, Vallée MP, Tao K, Roane TC, Williams GJ, Hopper JL, Southey MC, Andrulis IL, John EM, Goldgar DE, Lesueur F, Tavtigian SV. 2014.** Rare key functional domain missense substitutions in MRE11A, RAD50, and NBN contribute to breast cancer susceptibility: results from a Breast Cancer Family Registry case-control mutation-screening study. *Breast Cancer Research* **16(3)**:Article R58 DOI 10.1186/bcr3669.

**De Brakeleer S, De Grève J, Loris R, Janin N, Lissens W, Sermijn E, Teugels E. 2010.** Cancer predisposing missense and protein truncating BARD1 mutations in non-BRCA1 or BRCA2 breast cancer families. *Human Mutation* **31(3)**:E1175–E1185 DOI 10.1002/humu.21200.

**Eng C. PTEN Hamartoma Tumor Syndrome. 2016.** In: Pagon RA, Adam MP, Ardinger HH, Wallace SE, Amemiya A, Bean LJH, Bird TD, Ledbetter N, Mefford HC, Smith RJH, Stephens K, eds. GeneReviews®[Internet]. Seattle (WA): University of

Washington, Seattle; 1993–2017. *Available at http://www.ncbi.nlm.nih.gov/books/ NBK1488/*.

**Frantzen C, Klasson TD, Links TP, Giles RH, Von Hippel-Lindau Syndrome. 2015.** In: Pagon RA, Adam MP, Ardinger HH, Wallace SE, Amemiya A, Bean LJH, Bird TD, Ledbetter N, Mefford HC, Smith RJH, Stephens K, eds. GeneReviews [Internet]. Seattle (WA): University of Washington, Seattle; 1993–2017. *Available at http: //www.ncbi.nlm.nih.gov/books/NBK1463/*.

**Giusti F, Marini F, Brandi ML, Multiple Endocrine Neoplasia Type 1. 2015.** In: Pagon RA, Adam MP, Ardinger HH, Wallace SE, Amemiya A, Bean LJH, Bird TD, Ledbetter N, Mefford HC, Smith RJH, Stephens K, eds. GeneReviews [Internet]. Seattle (WA): University of Washington, Seattle; 1993–2017. *Available at http: //www.ncbi.nlm.nih.gov/books/NBK1538/*.

**Helgason H, Rafnar T, Olafsdottir HS, Jonasson JG, Sigurdsson A, Stacey SN, Jonasdottir A, Tryggvadottir L, Alexiusdottir K, Haraldsson A, Le Roux L, Gudmundsson J, Johannsdottir H, Oddsson A, Gylfason A, Magnusson OT, Masson G, Jonsson T, Skuladottir H, Gudbjartsson DF, Thorsteinsdottir U, Sulem P, Stefansson K. 2015.** Loss-of-function variants in ATM confer risk of gastric cancer. *Nature Genetics* **47**(**8**):906–910 DOI 10.1038/ng.3342.

**Hirotsu Y, Nakagomi H, Sakamoto I, Amemiya K, Oyama T, Mochizuki H, Omata M. 2015.** Multigene panel analysis identified germline mutations of DNA repair genes in breast and ovarian cancer. *Molecular Genetics and Genomic Medicine* **3**(**5**):459–466.

**Jasperson KW, Burt RW, *APC*-Associated Polyposis Conditions. 2014.** In: Pagon RA, Adam MP, Ardinger HH, Wallace SE, Amemiya A, Bean LJH, Bird TD, Ledbetter N, Mefford HC, Smith RJH, Stephens K, eds. GeneReviews [Internet]. Seattle (WA): University of Washington, Seattle; 1993–2017. *Available at http://www.ncbi.nlm.nih. gov/books/NBK1345/*.

**Kaurah P, Huntsman DG, Hereditary Diffuse Gastric Cancer. 2014.** In: Pagon RA, Adam MP, Ardinger HH, Wallace SE, Amemiya A, Bean LJH, Bird TD, Ledbetter N, Mefford HC, Smith RJH, Stephens K, eds. GeneReviews [Internet]. Seattle (WA): University of Washington, Seattle; 1993–2017. *Available at http://www.ncbi.nlm.nih. gov/books/NBK1139/*.

**Kirmani S, Young WF, Hereditary Paraganglioma-Pheochromocytoma Syndromes. 2014.** In: Pagon RA, Adam MP, Ardinger HH, Wallace SE, Amemiya A, Bean LJH, Bird TD, Ledbetter N, Mefford HC, Smith RJH, Stephens K, eds. GeneReviews [Internet]. Seattle (WA): University of Washington, Seattle; 1993–2017. *Available at http://www.ncbi.nlm.nih.gov/books/NBK1548/*.

**Klonowska K, Ratajska M, Czubak K, Kuzniacka A, Brozek I, Koczkowska M, Sniadecki M, Debniak J, Wydra D, Balut M, Stukan M, Zmienko A, Nowakowska B, Irminger-Finger I, Limon J, Kozlowski P. 2015.** Analysis of large mutations in BARD1 in patients with breast and/or ovarian cancer: the Polish population as an example. *Scientific Reports* **5**:Article 10424 DOI 10.1038/srep10424.

**Larsen Haidle J, Howe JR, Juvenile Polyposis Syndrome. 2015.** In: Pagon RA, Adam MP, Ardinger HH, Wallace SE, Amemiya A, Bean LJH, Bird TD, Ledbetter N,

Mefford HC, Smith RJH, Stephens K, eds. GeneReviews [Internet]. Seattle (WA): University of Washington, Seattle; 1993–2017. *Available at http://www.ncbi.nlm.nih. gov/books/NBK1469/*.

**Li J, Meeks H, Feng BJ, Healey S, Thorne H, Makunin I, Ellis J, Kconfab Investigators, Campbell I, Southey M, Mitchell G, Clouston D, Kirk J, Goldgar DI, Chenevix-Trench G. 2016.** Targeted massively parallel sequencing of a panel of putative breast cancer susceptibility genes in a large cohort of multiple-case breast and ovarian cancer families. *Journal of Medical Genetics* **53(1)**:34–42 DOI 10.1136/jmedgenet-2015-103452.

**Marquard J, Eng C. Multiple Endocrine Neoplasia Type 2. 2015.** In: Pagon RA, Adam MP, Ardinger HH, Wallace SE, Amemiya A, Bean LJH, Bird TD, Ledbetter N, Mefford HC, Smith RJH, Stephens K, eds. GeneReviews [Internet]. Seattle (WA): University of Washington, Seattle; 1993–2017. *Available at http://www.ncbi.nlm.nih. gov/books/NBK1257/*.

**National Cancer Institute. 2016.** PDQ genetics of skin cancer. Bethesda: National Cancer Institute. Date last modified 02/16/2016. *Available at http://www.cancer.gov/types/ skin/hp/skin-genetics-pdq* (accessed on 9 May 2016).

**National Comprehensive Cancer Network. 2016.** Gastric Cancer. Version 3.2016. *Available at https://www.nccn.org/professionals/physician_gls/pdf/gastric.pdf* (accessed on 30 August 2016).

**National Comprehensive Cancer Network. 2017.** Genetic/familial high risk assessment: breast and ovarian. Version 2.2017. *Available at https://www.nccn.org/professionals/ physician_gls/PDF/genetics_screening.pdf* (accessed on 12 January 2017).

**National Comprehensive Cancer Network. 2016.** Genetic/familial high risk assessment: colorectal. Version 1.2016. *Available at https://www.nccn.org/professionals/physician_ gls/pdf/genetics_colon.pdf* (accessed on 30 August 2016).

**National Comprehensive Cancer Network. 2016.** Neuroendocrine tumors. Version 2.2016. *Available at https://www.nccn.org/professionals/physician_gls/pdf/ neuroendocrine.pdf* (accessed on 30 August 2016).

**Ollier M, Radosevic-Robin N, Kwiatkowski F, Ponelle F, Viala S, Privat M, Uhrhammer N, Bernard-Gallon D, Penault-Llorca F, Bignon YJ, Bidet Y. 2015.** DNA repair genes implicated in triple negative familial non-BRCA1/2 breast cancer predisposition. *American Journal of Cancer Research 15;* **5(7)**:2113–2126.

**Potrony M, Badenas C, Aguilera P, Puig-Butille JA, Carrera C, Malvehy J, Puig S. 2015.** Update in genetic susceptibility in melanoma. *Annals of Translational Medicine* **3(15)**:Article 210 DOI 10.3978/j.issn.2305-5839.2015.08.11.

**Rafnar T, Gudbjartsson DF, Sulem P, Jonasdottir A, Sigurdsson A, Jonasdottir A, Besenbacher S, Lundin P, Stacey SN, Gudmundsson J, Magnusson OT, Le Roux L, Orlygsdottir G, Helgadottir HT, Johannsdottir H, Gylfason A, Tryggvadottir L, Jonasson JG, De Juan A, Ortega E, Ramon-Cajal JM, García-Prats MD, Mayordomo C, Panadero A, Rivera F, Aben KK, Van Altena AM, Massuger LF, Aavikko M, Kujala PM, Staff S, Aaltonen LA, Olafsdottir K, Bjornsson J, Kong A, Salvarsdottir A, Saemundsson H, Olafsson K, Benediktsdottir KR, Gulcher**

J, Masson G, Kiemeney LA, Mayordomo JI, Thorsteinsdottir U, Stefansson K. **2011.** Mutations in BRIP1 confer high risk of ovarian cancer. *Nature Genetics* **43(11)**:1104–1107 DOI 10.1038/ng.955.

Roberts NJ, Jiao Y, Yu J, Kopelovich L, Petersen GM, Bondy ML, Gallinger S, Schwartz AG, Syngal S, Cote ML, Axilbund J, Schulick R, Ali SZ, Eshleman JR, Velculescu VE, Goggins M, Vogelstein B, Papadopoulos N, Hruban RH, Kinzler KW, Klein AP. **2012.** ATM mutations in hereditary pancreatic cancer patients. *Cancer Discovery* **2**:41–46 DOI 10.1158/2159-8290.CD-11-0194.

Schneider K, Zelley K, Nichols KE, Garber J, Li-Fraumeni Syndrome. **1999.** In: Pagon RA, Adam MP, Ardinger HH, Wallace SE, Amemiya A, Bean LJH, Bird TD, Ledbetter N, Mefford HC, Smith RJH, Stephens K, eds. GeneReviews [Internet]. Seattle (WA): University of Washington, Seattle; 1993–2017. *Available at http://www.ncbi.nlm.nih.gov/books/NBK1311/*.

Seemanová E, Jarolim P, Seeman P, Varon R, Digweed M, Swift M, Sperling K. **2007.** Cancer risk of heterozygotes with the NBN founder mutation. *Journal of the National Cancer Institute* **99(24)**:1875–1880 DOI 10.1093/jnci/djm251.

Soura E, Eliades PJ, Shannon K, Stratigos AJ, Tsao H. **2016.** Hereditary melanoma: Update on syndromes and management. Genetics of familial atypical multiple mole melanoma syndrome. *Journal of the American Academy of Dermatology* **74**:395–407 DOI 10.1016/j.jaad.2015.08.038.

Zhang G, Zeng Y, Liu Z, Wei W. **2013.** Significant association between Nijmegen breakage syndrome 1 657del5 polymorphism and breast cancer risk. *Tumor Biology* **34(5)**:2753–2757 DOI 10.1007/s13277-013-0830-z.