# Peer review of "Development and validation of a 36-gene sequencing assay for hereditary cancer risk assessment"

_PeerJ, doi:10.7717/peerj.3046_

## Round 0.1 · original submission · Major Revisions

Please address the reviewer comments thoroughly.

Reviewer 1 ·

Basic reporting

The manuscript is well written and clearly laid out with appropriate headings, references and labeling. Table S6 has one typo - should be ‘visual’. The background detail is appropriate and proportionate. The tables are all relevant and appropriately laid out with appropriate headings. The discussion is well written and the conclusions appropriate for the work presented

Experimental design

The rationale for undertaking the study has been explained and is entirely appropriate as is the experimental design. The study as been conducted to a sufficiently high standard and does fill a knowledge gap – incorporating CNV detection into NGS panel testing.

Validity of the findings

The research design is appropriate and rigorous. Sufficient consideration has been given to the issue of false positive and false negative findings and appropriate controls have been used. I am, however, not sufficiently qualified to comment on the statistical analysis, in particular the analyses used in the CNV Calling Algorithm and this section would benefit from the input of a more expert reviewer in this particular area. However, I do think it extremely helpful from the clinical perspective that the laboratory has sought to incorporate CNV detection into their pipeline along with indels.

Regarding validity, I do have the following points to raise:

1. Was a MUTYH positive control available for analysis, or are the authors certain they can pick up biallelic mutations of MUTYH given its recessive nature?

2. I appreciate the remit of this manuscript is the laboratory analysis to a clinical standard of the 36 genes on the panel and the content of the panel is outside of this being up to the referring clinician to decide what genes ought to be investigated. However, whilst the core genes are present, for example BRCA1, BRCA2, MMR genes, it is notable that rare genes such as VHL are present, but not other renal cancer predisposing genes such as FH and FLCN (and most likely as common, if not moreso). Also SDHA for phaeochromocytoma / paraganglioma predisposition is on the panel (rare), but SDHD is not. There are a few other examples. So I am not sure which conditions the panel is aimed toward, my feeling is hereditary breast / ovarian / bowel cancer with a few extras added on. The discussion would benefit from a sentence re the clinical utility of the panel.

Reviewer 2 ·

Basic reporting

This is generally a well written paper worthy of publication. It is not particularly cutting edge as it relates to assay validation but is nevertheless valuable in that it demonstrate a thorough validation of a clinical grade assay and seems to demonstrate a high sensitivity and specificity that can be achieved with a well validated test. In my opinion more evidence needs to be provided to back up the assertions of high sensitivity and specificity for the assay within the parameters that the authors describe.

Experimental design

No comment

Validity of the findings

There is a lack of supporting evidence for some of the conclusions. The most important lack of supporting evidence relates to the minimum sequence coverage depth that the authors state that the assay has to meet in order for analysis to be successful.
The authors state on lines 127-128 ‘All target nucleotides are required to be covered with a minimum depth of 20 reads.’
Read depth of 20 is low for heterozygous mutation detection (and probably too low in my opinion for reliable heterozygous mutation detection). Heterozygote variant detection efficiency is adversely affected by low coverage depth. Published data has shown that coverage depth <30x is inadequate for reliable heterozygote mutation detection (REF: Ajay S et al (2011). Accurate and comprehensive sequencing of personal genomes. Genome Research 21;1498-1505.) Clearly improvement in bioinformatic pipelines and sequencing quality has occurred since 2011 however, it is not possible to tell from the data given that the authors have demonstrated that their bioinformatics pipeline is capable of achieving the stated sensitivities at this low coverage depth. The authors should be required to provide evidence that mutations of all classes SNVs, indels and CNVs are detected reliably at their minimum stated read depth. This data can be generated by downsampling (analysis on bam files with randomly removed reads) if necessary.
It is also not clear from the data provided what the total number of unique variants assessed during the validation was. It is clear from the supplementary tables e.g. Table S4a that many of the variants detected are duplicates. It is self-evident that once a given variant is detected it is more likely to be detectable in a different sample than a completely different variant. In order for the reader to judge the true sensitivity of the assay the total number and range of variants assessed should be listed in a Supplementary table (including SNVs, indels and CNVs). In my opinion (and of others) the sensitivity and specificity measures (and confidence intervals) should be based on the number of unique variants detected, being able to detect the same mutation/variant in a different sample is a measure of reproducibility (REF: Mattocks et al (2010) A standardized framework for the validation and verification of clinical molecular genetic tests. European Journal of Human Genetics 18, 1276–1288)

The authors state on lines 187-188 ‘All copy-number called variants are inspected for quality of raw data by human review, and 187 observed positive variants are rerun in our production SOP for verification of the call.’ What does this mean? Do the authors re-run the data through the same pipeline again or is this validation with an orthogonal method?

The authors state that CNV detection was determined on a 44 sample set. Were CNVs not tested for in the remaining samples? If not why was this NGS data not analysed for CNVs?

On line 338 the authors refer to Table S6a, I could not find this Table in the supplementary information supplied, should this actually refer to Table S4a?
It is not clear from the manuscript whether operator and instruments were varied in the inter-run reproducibility experiments. This should be stated.

Reviewer 3 ·

Basic reporting

This manuscript reports the technical capability of a commercial laboratory to genetic sequence variant detection. The process, both wet lab and bioinformatics plus analytical validation processes are clearly described and form the main point of the manuscript. The point that sensitivity and specificity are a function of different NGS pipelines and each needs to be evaluated so that limitations are understood.
There is no particular hypothesis.

Experimental design

Samples – it would be more relevant for a panel of hereditary cancer predisposition genes if the test sensitivity had been assessed in FFPE than in saliva since the main problem in identifying cancer familial predisposition is no living affected relative from whom to take a DNA sample.

The methods are clear and a logical approach to validation is presented.

Validity of the findings

Hereditary cancer genes
The genes included in the 36-gene panel are somewhat irrelevant since there are no data presented in the paper that validate or justify clinically the inclusion of the genes that are described other than reference to other literature.
Hereditary cancer - clinical utility of testing a panel of genes
It is not clear that extending the same scrutiny or intervention to low and moderate penetrance genes will bring the same projected benefits in risk reduction with considerable potential for harm if results or risks are misrepresented or misinterpreted. It is not reasonable to base general screening recommendations on anecdotal case reports.

Additional comments

Overall a clearly written and useful technical account of a well developed approach to NGS although the relevance to the disease area is somewhat incidental. It would be helpful to emphasise that not all genes are clinically equivalent and cautious reporting of clinical assertions for variants in less well characterised genes is important to consider when a broad range of genes is routinely analysed. The workload associated with assessment of variants across a very broad cancer gene panel is significant with in most situations a vary marginal uplift in "diagnosis" (Maxwell et al The American Journal of Human Genetics 98, 801–817, May 5, 2016).

---

## Round 0.2 · accepted · Accept

You have responded satisfactorily to the reviewer comments.